# New Insights into the Development of Donepezil-Based Hybrid and Natural Molecules as Multi-Target Drug Agents for Alzheimer’s Disease Treatment

**DOI:** 10.3390/molecules29225314

**Published:** 2024-11-11

**Authors:** Violina T. Angelova, Boris P. Stoyanov, Rumyana Simeonova

**Affiliations:** 1Department of Chemistry, Faculty of Pharmacy, Medical University of Sofia, 1000 Sofia, Bulgaria; 2Department of Pharmacology, Pharmacotherapy and Toxicology, Faculty of Pharmacy, Medical University of Sofia, 1000 Sofia, Bulgaria; bobi.stoyanov@abv.bg

**Keywords:** Alzheimer’s disease, antioxidation, beta-amyloid, benzylpiperidine hybrids, cholinergic, donepezil analogs, multi-target drugs, melatonin, tau hyperphosphorylation, neuroinflammation, natural molecules

## Abstract

Alzheimer’s disease (AD) involves a complex pathophysiology with multiple interconnected subpathologies, including protein aggregation, impaired neurotransmission, oxidative stress, and microglia-mediated neuroinflammation. Current treatments, which generally target a single subpathology, have failed to modify the disease’s progression, providing only temporary symptom relief. Multi-target drugs (MTDs) address several subpathologies, including impaired aggregation of pathological proteins. In this review, we cover hybrid molecules published between 2014 and 2024. We offer an overview of the strategies employed in drug design and approaches that have led to notable improvements and reduced hepatotoxicity. Our aim is to offer insights into the potential development of new Alzheimer’s disease drugs. This overview highlights the potential of multi-target drugs featuring heterocycles with *N*-benzylpiperidine fragments and natural compounds in improving Alzheimer’s disease treatment.

## 1. Introduction

In recent years, there has been a growing interest in the multi-target and polypharmacologic approach to treating various diseases, including Alzheimer’s disease (AD), to develop new, more effective, and selective drugs that have fewer side effects and can address the emergence of drug resistance. Combining two or more drugs in clinical practice has shown promising therapeutic outcomes [1,2]. Additionally, co-formulations are used and, more notably, hybrid or chimeric molecules that can target multiple pathways involved in Alzheimer’s disease have been developed. As of 2024, around 55 million people worldwide are living with Alzheimer’s and other forms of dementia—a number expected to rise to 139 million by 2050 due to aging populations [2]. This alarming increase highlights the urgent need for effective treatments to manage and potentially slow the disease’s progression. Currently, 156 clinical trials are exploring brain changes, such as tau protein accumulation and inflammation, as potential therapeutic targets for Alzheimer’s disease. Current treatment options, such as aducanumab, lecanemab, and donanemab, which target beta-amyloid plaques, represent significant advances but come with limitations. Aducanumab is being discontinued, while lecanemab and donanemab have shown moderate benefits in the early stages of Alzheimer’s, which includes the mild cognitive impairment (MCI) or mild dementia stage of the disease [3]. However, these drugs require intravenous infusions and careful monitoring for amyloid-related imaging abnormalities (ARIAs), a serious potential side effect. While drugs like aducanumab, lecanemab, and donanemab offer hope, they are part of a broader and evolving landscape of Alzheimer’s research [4]. The majority of Alzheimer’s drugs, including donepezil, rivastigmine, galantamine, memantine, and a memantine-donepezil combination, focus on treating cognitive symptoms without altering disease progression [5]. These medications can cause side effects like nausea, diarrhea, dizziness, headaches, and, in some cases, mood swings or behavioral changes [6]. Additionally, cholinesterase inhibitors may interact with other medications that affect heart rate or blood pressure, necessitating careful management [7,8]. Given the rising prevalence of Alzheimer’s, there is an increasing need for treatments that can effectively manage the disease while minimizing side effects. Multi-target drugs offer a promising approach by addressing the complex and multifaceted nature of Alzheimer’s [9]. These drugs, which can target multiple mechanisms involved in the disease, have the potential to provide more effective and comprehensive treatment options. In medicinal chemistry, particularly for diseases like Alzheimer’s, advancements in conjugate chemistry methods present a promising pathway for enhancing drug quality [10]. By linking known pharmacophores to create new molecules with improved properties, researchers can potentially develop more potent and diverse treatment options, better equipped to address the challenges of Alzheimer’s and other complex diseases.

## 2. Current Drugs for Treatments

As of now, eight drugs are commonly used to treat Alzheimer’s disease (Figure 1) [11]. These medications fall into three categories: cholinesterase inhibitors, NMDA receptor antagonists, and those targeting and removing beta-amyloid plaques. Additionally, they are classified as (a) medications for mild-to-moderate Alzheimer’s disease; (b) medications for moderate-to-severe Alzheimer’s disease; and (c) medications to be used with caution in people with Alzheimer’s disease.

### 2.1. Treatments Focused on Slowing Alzheimer’s Progression: Aducanumab, Lecanemab, and Donanemab

Aducanumab, lecanemab, and donanemab are FDA-approved drugs designed to slow the progression of Alzheimer’s disease by targeting and removing beta-amyloid plaques in the brain, a key factor believed to contribute to the condition (Figure 2) [3]. Aducanumab is a monoclonal antibody that targets these amyloid-beta plaques. Approved by the FDA in June 2021, aducanumab has been the subject of significant controversy regarding its effectiveness [12]. Although it is administered via intravenous infusion and has the potential to slow Alzheimer’s progression, aducanumab is being discontinued for resource reasons. However, it will remain available to existing patients until November 2024 [13]. Lecanemab is another monoclonal antibody targeting amyloid-beta. Approved by the FDA in 2023, lecanemab has shown moderate benefits in the early stages of Alzheimer’s, particularly in individuals with mild cognitive impairment [14]. Like aducanumab, lecanemab is given intravenously and carries the risk of amyloid-related imaging abnormalities (ARIAs), temporary brain swelling that requires careful monitoring [15]. Aducanumab and lecanemab present promising options for slowing the progression of Alzheimer’s disease, yet ongoing research is crucial for developing more comprehensive treatments and potentially altering the disease’s course. Donanemab is an investigational monoclonal antibody that targets amyloid-beta plaques in the brain, which are associated with Alzheimer’s disease (AD). It has shown potential for slowing cognitive decline in patients with early-stage AD, particularly those with lower levels of tau protein, which is linked to disease severity. Donanemab’s mechanism involves binding to a modified form of amyloid, promoting plaque clearance and potentially altering the course of the disease [3]. While these drugs mark significant advancements in Alzheimer’s therapy, they are not cures and may not be effective for all patients. Early diagnosis remains vital for selecting the most appropriate treatment and managing potential risks, such as amyloid-related imaging abnormalities (ARIAs), effectively. 

### 2.2. Treatments to Address Cognitive and Behavioral Symptoms—Donepezil, Rivastigmine, Galantamine, Memantine, and a Memantine-Donepezil Combo

Five of the eight approved Alzheimer’s drugs—donepezil, rivastigmine, galantamine, memantine, and the memantine-donepezil combination—target cognitive symptoms without affecting disease progression (Figure 2) [16]. Donepezil, rivastigmine, and galantamine are cholinesterase inhibitors that enhance acetylcholine levels, while memantine is an NMDA receptor antagonist that regulates glutamate activity. These medications can cause side effects like headache and nausea [17]. Memantine, approved in 2003 for moderate-to-severe Alzheimer’s disease, is a noncompetitive NMDA receptor antagonist that improves neural signaling and prevents excessive calcium entry into neurons, offering neuroprotection [18]. Clinical studies show it has minimal liver side effects, even when combined with cholinesterase inhibitors. In 2014, the FDA approved a memantine-donepezil combination, which has shown superior results in enhancing cognitive function and overall patient condition compared to donepezil alone [19].

Brexpiprazole is an atypical antipsychotic used to treat major depressive disorder (MDD), schizophrenia, and agitation associated with Alzheimer’s disease dementia. It acts as a serotonin-dopamine activity modulator (SDAM), though its exact mechanism is unclear.

While effective as adjunct therapy for MDD and in treating schizophrenia in adults and children 13+, it carries risks such as stroke and increased mortality in elderly dementia patients. Common side effects include weight gain, drowsiness, akathisia, dizziness, and nasopharyngitis. Due to these risks, non-pharmacological treatments should be prioritized. Additionally, warnings include the potential for neuroleptic malignant syndrome, tardive dyskinesia, seizures, metabolic changes, and compulsive behaviors.

In 2019, sodium oligomannate (GV-971) received conditional approval in China to improve cognitive function in mild-to-moderate Alzheimer’s disease (AD) by targeting the brain–gut axis. Derived from marine algae, sodium oligomannate reshapes gut microbiota, reducing neuroinflammation, amyloid-beta (Aβ) accumulation, and tau protein hyperphosphorylation. This novel approach highlights the gut–brain axis’s role in AD. Research shows gut microbiota dysbiosis can trigger inflammation and contribute to AD progression. Sodium oligomannate reduces gut-related inflammation, improving cognitive function. While promising, further research is needed to confirm its long-term efficacy and safety. Suvorexant, approved for insomnia, has shown effectiveness in managing sleep issues in mild-to-moderate Alzheimer’s cases by inhibiting orexin, a neurotransmitter involved in the sleep–wake cycle. However, it may cause side effects such as impaired alertness, motor coordination issues, worsened depression, sleep behaviors like sleep-walking, and reduced respiratory function.

## 3. Adjuvants of Multi-Target Drugs in Alzheimer’s

Multi-target drugs, or multi-target-directed ligands (MTDLs), offer several significant advantages in the treatment of Alzheimer’s disease due to their ability to interact with multiple biological targets simultaneously. Their advantages include the following:

Comprehensive disease modulation: These drugs offer a multi-targeted approach to Alzheimer’s, addressing several pathological mechanisms such as amyloid-beta accumulation, tau protein phosphorylation, neuroinflammation, and oxidative stress. By impacting these interconnected pathways, they provide a more integrated and holistic strategy for disease management, which may improve clinical outcomes and address the disease’s complexity more effectively [20,21,22,23,24].

Improved efficacy: Multi-target drugs hold promise for increased effectiveness by addressing several Alzheimer’s pathways simultaneously. For example, a drug designed to inhibit amyloid-beta production and to reduce tau aggregation could more comprehensively limit the buildup of pathological proteins, potentially slowing disease progression more effectively than single-target treatments. This combined approach may help tackle the disease’s multifactorial nature and improve clinical outcomes by reducing both amyloid and tau pathology [25,26,27,28,29,30].

Reduction in disease progression: By simultaneously targeting multiple pathological mechanisms, multi-target drugs have the potential to slow Alzheimer’s disease progression more effectively than single-target treatments. This approach could help to reduce amyloid and tau accumulation, control neuroinflammation, and counteract oxidative stress, leading to improved overall outcomes and delaying the onset of severe symptoms. This broad-based strategy addresses the interconnected nature of Alzheimer’s pathology, potentially offering a more sustained impact on disease advancement and a better quality of life for patients [27,28,31,32,33,34,35,36].

Lower risk of drug resistance: By targeting multiple pathways, multi-target drugs can reduce the likelihood of Alzheimer’s disease developing resistance to treatment, a crucial advantage in managing complex diseases. This approach not only tackles various aspects of the disease but also lowers the chance that the brain’s pathology can adapt to a single mechanism, thereby sustaining treatment efficacy over time. Multi-target drugs offer a more robust approach to prevent compensatory mechanisms that might otherwise undermine single-target therapies, enhancing long-term treatment success and improving patient outcomes [37,38,39,40].

Enhanced patient compliance: A single multi-target drug addressing various Alzheimer’s mechanisms can streamline treatment by reducing the need for multiple medications, which may improve patient adherence. Simplifying complex treatment regimens into one medication minimizes pill burden, lowers the risk of missed doses, and reduces potential drug interactions. This can be particularly beneficial for Alzheimer’s patients who may struggle with memory and organization, ultimately supporting more consistent therapeutic outcomes and reducing caregiver burden [33,41,42,43,44].

Potential for personalized therapy: Multi-target drugs offer the possibility of customization based on each patient’s specific disease characteristics and progression. By targeting specific pathways like amyloid-beta, tau, neuroinflammation, or oxidative stress, these drugs can be tailored to align with individual biomarker profiles or genetic predispositions [28,33,45,46,47,48,49].

Reduction in side effects: Administering a single multi-target drug may allow for lower doses than those required with multiple single-target drugs, potentially reducing the overall risk of adverse effects. By addressing several disease mechanisms within one compound, patients may avoid the cumulative side effects that can arise from polypharmacy, particularly in elderly populations who are more susceptible to drug interactions. This approach not only improves the safety profile of Alzheimer’s treatments but also enhances patient comfort and adherence, contributing to better therapeutic outcomes over time [50,51,52,53,54].

Streamlined drug development: Multi-target drugs can simplify therapeutic regimens, potentially reducing both the complexity and cost of drug development and administration. By combining mechanisms of action within a single compound, these drugs may streamline clinical trial processes and reduce the need for separate studies on multiple single-target drugs. This consolidation also has the potential to lower production and regulatory costs, ultimately benefiting healthcare systems and patients alike. A unified multi-target approach reduces the resource demand typical of developing multiple single-target drugs, potentially accelerating time-to-market and enhancing access to effective Alzheimer’s treatments.

Potential for disease modification: Beyond merely managing symptoms, multi-target drugs may have the potential to modify the underlying processes of Alzheimer’s disease. By addressing multiple pathological mechanisms, such as amyloid-beta accumulation, tau hyperphosphorylation, and neuroinflammation, these drugs can potentially lead to more significant improvements in disease progression and enhance patient quality of life [50,55].

Innovative therapeutic strategies: The use of multi-target drugs in Alzheimer’s treatment represents a groundbreaking approach that may lead to significant breakthroughs in disease management and our understanding of the disease. By simultaneously addressing various pathological mechanisms, these therapies offer a novel way to tackle the complexity of Alzheimer’s [56,57].

In summary, as monotherapy for complex diseases becomes less effective due to resistance and side effects, multi-target drug strategies are emerging as a promising alternative. The concept of polypharmacology is gaining traction among researchers and pharmaceutical companies, who are increasingly focused on developing these drugs. By employing computational methods to screen protein networks and identify key interactions, researchers can design multi-target drugs that may better address resistance. Choosing the optimal combination of targets for both multi-target drugs and therapeutic combinations remains challenging. It requires a thorough understanding of target–disease associations, pathway–target–drug–disease interactions, and adverse event profiling. Additionally, the selection process should consider whether modulating the chosen targets will result in additive or synergistic effects. Additive effects occur when the targets are part of the same pathway, while synergistic effects arise from targets in functionally complementary pathways. Both scenarios typically allow for lower doses and potentially better safety profiles compared to single-target drugs. Consequently, the quest for multi-target medications is likely to continue, potentially offering our best chance for developing effective treatments against complex diseases such as Alzheimer’s disease.

The complexity of AD’s molecular pathogenesis necessitates a multifaceted treatment approach. Multi-target drug design aligns well with this need, as it allows for simultaneous intervention in various disease mechanisms, offering a potentially more effective and comprehensive therapeutic strategy.

## 4. Therapeutic Strategies for Alzheimer’s Disease (AD)

As mentioned above, the pathogenesis of Alzheimer’s disease (AD) involves amyloid-beta (Aβ) plaques, neurofibrillary tangles (NFTs), synapse loss, oxidative stress, and neuronal death. Initial research focused on acetylcholinesterase (AChE) and butyrylcholinesterase (BChE) enzymes, targeting dual inhibition for potential therapeutic benefits [58,59,60]. Additionally, factors such as synapse loss, oxidative stress, and neuronal death are also implicated and often occur alongside these primary markers [61,62,63,64]. The amyloid cascade involves Aβ peptide aggregation, disrupting cellular functions and increasing neurotoxicity [65,66,67,68,69,70]. Other factors include the blood–brain barrier’s role [71,72,73] in drug delivery, oxidative stress [58,74,75,76,77], neuroinflammation [78,79,80,81], and calcium signaling disruptions [82,83,84].

Given the multifactorial nature of AD, single-target treatments have shown limited success, underscoring the need for hybrid multi-target compounds. These hybrids are designed to address multiple AD pathways, such as reducing Aβ aggregation, inhibiting tau hyperphosphorylation, enhancing antioxidant defenses, and modulating neurotransmitter systems. By targeting various mechanisms, they offer a more comprehensive treatment approach with potentially improved therapeutic outcomes [85,86], as presented in Figure 3.

## 5. Novel Donepezil-Based Hybrids with a Focus on *N*-Benzylpiperidine Derivatives for Targeting AD (2014–2024)

Heterocyclic compounds are key in multi-target-directed ligands’ (MTDLs’) development, requiring precise chemical synthesis to ensure efficacy and safety. The goal is to better address the complex nature of neurodegenerative diseases, potentially offering advantages over single-target therapies. Here, we examine advancements in donepezil-based hybrids, focusing on the design, synthesis, and evaluation of *N*-benzylpiperidine derivatives for Alzheimer’s disease treatment over the past decade. The piperidine moiety can interact with various biological targets, making it a valuable scaffold in medicinal chemistry. *N*-Benzyl substitution increases lipophilicity, enhancing membrane penetration and bioavailability. The nitrogen atom in the piperidine ring adds electron density, influencing binding affinity to neuropharmacological receptors and enzymes. Additionally, the structure allows conformational flexibility, enabling optimal interactions with target proteins. This versatility aids in designing new derivatives with improved efficacy and selectivity while reducing side effects. The nitrogen atom can also participate in hydrogen bonding, further enhancing binding affinity to specific targets. This review also highlights the potential of various natural compounds as multi-targeted therapies for Alzheimer’s disease (AD), addressing multiple disease pathways.

Acetylcholinesterase inhibitors (AChEIs) enhance cholinergic neurotransmission in the brain by increasing endogenous acetylcholine levels. One of the most effective and well-known FDA-approved AChEIs is donepezil (DP) (Figure 4) (2-((1-benzylpiperidin-4-yl)methyl)-5,6-dimethoxy-2,3-dihydro-1*H*-inden-1-one), which features a dimethoxy indanone structure connected to *N*-benzylpiperidine [87] via a methylene linker. DP not only inhibits AChE but also exhibits anti-amyloid-beta (Aβ) aggregation, antioxidant, and metal-chelating activities. Modifying the dimethoxy indanone or *N*-benzylpiperidine with different heterocyclic scaffolds has led to the development of new donepezil hybrids, each showing distinct inhibitory properties [87,88,89]. We emphasize the role of the *N*-benzylpiperidine (*N*-BP) motif in drug discovery efforts due to its structural flexibility and three-dimensional configuration. Medicinal chemists often employ the *N*-BP motif as a versatile element to refine both the therapeutic efficacy and physicochemical properties of drug candidates. It plays a key role in forming cation–π interactions with target proteins and serves as a foundation for optimizing stereochemistry, which can influence both potency and toxicity. This motif is present in many approved drugs as well as compounds in clinical or preclinical development (Table 1). In addition to its use in AChE inhibitors, the *N*-BP motif has been incorporated into multi-target-directed ligands (MTDLs) aimed at addressing several pathological mechanisms of Alzheimer’s simultaneously. By combining the *N*-BP motif with other pharmacophores, researchers design molecules that can not only inhibit AChE but also interact with other targets, such as β-amyloid plaques and tau tangles, which are hallmarks of Alzheimer’s disease [87].

Here, we focus mainly on well-established and incipient AD therapeutic targets, AChE, BuChE, MAOs, β-amyloid deposition, 5-HT4, and serotonin transporter, intending to shed light on new insights in AD multi-target therapy.

Banoo, R et al. (2024) [90] report the synthesis and biological evaluation of indole-piperidine amides as multi-target-directed ligands (MTDLs) for AD, identifying 5,6-dimethoxy-indole *N*-(2-(1-benzylpiperidine) carboxamide (**1**) as a dual AChE/BACE-1 (β-secretase) inhibitor (Table 1). Compound (**1**) (Table 1) is a mixed-type inhibitor with Ki values of 0.26 and 0.46 μM, respectively, and demonstrates excellent BBB permeability in the PAMPA assay. These in vitro results suggest that this compound warrants further investigation in animal models for in vivo efficacy. Molecular dynamics simulations also revealed that AChE and BACE-1 undergo minor conformational changes when binding to compound (**1**) (Table 1).

A series of indanone/benzofuranone and piperidine hybrids was designed and synthesized by Zeng, Q. et al. (2024) [91] based on the neuroprotective effects of butylphthalide and donepezil hybrids to improve the bioavailability and therapeutic efficacy of natural phthalide analogs. Most indanone derivatives with 1-methylpiperidine (**2**) (Table 1) in the tail segment showed stronger neuroprotective effects in an oxygen-glucose deprivation/reoxygenation (OGD/R)-induced neuronal injury model compared to benzofuranones. Among them, compound (**2**) (Table 1) displayed significant neuroprotection without cytotoxicity and excellent BBB permeability. In vivo studies showed that compound (**2**) reduced ischemia-reperfusion injury, lowering infarct volume to 18.45% at 40 mg/kg, outperforming edaravone at 20 mg/kg, suggesting its therapeutic potential for neurological disorders.

Zhai, J. (2024) [92] developed a dual-target inhibitor by combining the chemical structures of baicalein and donepezil (Table 1). The modification of baicalein into arylcoumarin led to the synthesis of three structural compounds, with compound (**3b**) showing the strongest AChE inhibition (IC50  =  0.05  ±  0.02 µM), outperforming both donepezil and baicalein. Compound (**3b**) also effectively inhibits Aβ1-42 aggregation, protects nerve cells, and penetrates the blood–brain barrier. In a zebrafish behavioral test, it alleviated movement retardation caused by AlCl3, making it a promising multifunctional agent for treating and managing AD symptoms.

In a study by Mohammadi-Farani, A. (2024) [93], a new series of benzamide derivatives was designed, synthesized, and characterized. Acetylcholinesterase inhibition was evaluated using Ellman’s method, and results were compared to donepezil (Table 1). Compound (**4**) was the most potent (IC50 = 0.14 ± 0.03 nM), surpassing donepezil. Molecular docking showed that (**4**) bound to AChE’s active site via a hydrogen bond with Trp279. This compound shows promise as a lead candidate, though further experimental in vivo testing is necessary to confirm its drug potential.

Butyrylcholinesterase (BChE) has become a critical target in Alzheimer’s disease (AD) research due to its role in acetylcholine (ACh) hydrolysis and its link to β-amyloid (Aβ) deposition, which worsens disease progression. In a study by Chen, Y., et al. (2024) [94], compound (**5**), a selective and potent BChE inhibitor (eqBChE IC50 = 0.059 ± 0.006 μM, hBChE IC50 = 0.162 ± 0.069 μM), was identified through virtual filtering and structural modification (Table 1). The compound exhibited excellent drug-like properties, including high oral bioavailability, metabolic stability, and blood–brain barrier (BBB) permeability, making it well suited for targeting the central nervous system (CNS). Compound (**5**) effectively protected neural cells from oxidative stress and inflammation in vitro and demonstrated promising in vivo results, improving cognition and reducing inflammation in mouse models induced by Aβ1-42 and lipopolysaccharides (LPS). It also reduced Aβ1-42 and inflammatory markers while increasing ACh levels, thereby preserving the neural microenvironment and alleviating cognitive symptoms. Overall, compound (**5**)’s neuroprotective and cognition-enhancing effects position it as a promising candidate for further research in AD treatment.

In a previous study [95], we explored two series of hybrid molecules combining melatonin and donepezil with hydrazone or sulfonyl hydrazone fragments. Lead compound (**6a**) exhibited significant AChE inhibition (10.76 ± 1.66 μM) and BChE inhibition (26.32 ± 3.11 μM), along with notable antioxidant activity and lipid peroxidation inhibition. Compound (**6b**) showed selective BChE inhibition (21.12 ± 1.48 μM; SI BChE = 47.34) and effectively prevented oxidative stress in SH-SY5Y cells. In antioxidant tests, compound **6a** demonstrated high DPPH activity and showed the best FRAP and FTC activity. These compounds exhibited low cytotoxicity, high bioavailability, and good BBB permeability. Molecular docking suggested that **6a** binds to MT1 and MT2 receptors, AChE, and BChE, making **6a** and **6b** promising candidates for AD treatment. In further studies, we (Tchekalarova, J., et al. 2024) [95,97] evaluated compound **6a** in AD and melatonin deficiency models, as well as the effects of lead compounds against Aβ-induced neurotoxicity and memory deficits in mice [96].

The work of Walker, D. K. et al. (2023) [98] presents a new class of compounds designed using a multi-targeted ligand approach for AD. They tested these compounds for in vitro inhibition of human acetylcholinesterase (hAChE), butyrylcholinesterase (hBChE), β-secretase-1 (hBACE-1), and amyloid β (Aβ) aggregation. Compounds (**7a**) and (**7b**) showed hAChE and hBACE-1 inhibition similar to donepezil and hBChE inhibition comparable to rivastigmine. They significantly reduced Aβ aggregation and showed no neurotoxic effects in SH-SY5Y cells. In AD mouse models, (**7a**) and (**7b**) improved learning and memory, reduced AChE, malondialdehyde, and nitric oxide levels, increased glutathione, and lowered pro-inflammatory cytokines. Histopathological and Western blot analyses showed normal brain structure and reduced Aβ, APP/Aβ, BACE-1, and tau protein levels. Compounds (**7a**) and (**7b**) are promising new leads for AD therapeutics.

Qin, P.J., et al. (2023) [99] designed, synthesized, and evaluated a series of *N*-benzyl piperidine derivatives for dual inhibition of histone deacetylase (HDAC) and acetylcholinesterase (AChE). Among the compounds tested, (**8a**) and (**8b**) demonstrated significant dual enzyme inhibition (**8a**): HDAC IC50 = 0.17 μM, AChE IC50 = 6.89 μM; (**8b**): HDAC IC50 = 0.45 μM, AChE IC50 = 3.22 μM). Histone deacetylases (HDACs) have recently gained attention as a promising target for AD treatment. Research in aged animal models has shown that reduced histone acetylation leads to the downregulation of genes essential for learning and memory, particularly in the hippocampus and cerebral cortex regions of the brain. Both compounds also exhibited free radical scavenging, metal chelation, and Aβ aggregation inhibition activities. Notably, (**8a**) and (**8b**) showed promising neuroprotective effects in PC-12 cells and good selectivity for AChE. These multifunctional properties highlight the potential of (**8a**) and (**8b**) for further optimization as treatments for AD.

Two primary scaffolds, pyrazolopyridine and tetrahydroacridine (THA), were employed from Waly, O. M (2022) [100] to develop four series of MTDLs targeting ChE (hAChE or hBuChE) and Aβ1-42 aggregation, along with optimal metal chelation properties. Structural modifications were made to the 9-amino group of the THA core of tacrine and pyrazolopyridine, linking them to various cyclic secondary amines using amide spacers, ethylamine bridges, or combining THA with pyrazolopyridine to create hybrid compounds. Different 9-amino substitutions improved the in vitro hAChE activity of 7- or 6,7-disubstituted THA derivatives. Compound (**9**) emerged as potent multimodal anti-AD agent, effectively inhibiting hAChE and binding to the peripheral anionic site (PAS), impacting Aβ aggregation and neurotoxicity. Notably, compound (**9**) was nearly twice as effective as donepezil. Compound (**9**) also inhibited Aβ1-42 self-aggregation and chelated bio-metals like Fe^2+^, Zn^2+^, and Cu^2+^, preventing reactive oxygen species (ROS) generation and oxidative brain damage. Compound **9**, with dual ChE activity, exhibited superior cognitive benefits. The compound demonstrated safety in hepG2 cells, excellent blood–brain barrier (BBB) penetration, and a wide safety margin, with LD50 values exceeding 120 mg/kg.

A series of 36 new *N*-alkylpiperidine carbamates was developed by Košak, U. (2020) [101] as potential anti-Alzheimer’s agents targeting cholinesterases (AChE, BChE) and monoamine oxidases (MAO-A, MAO-B). Two compounds showed promise: compound (**10**) inhibited AChE (IC50 = 7.31 μM), BChE (IC50 = 0.56 μM), and MAO-B (IC50 = 26.1 μM); and compound (**11**) selectively inhibited MAO-B (IC50 = 0.18 μM). Compounds (**10**) and (**11**) (Table 1) can cross the blood–brain barrier and are non-cytotoxic. Compounds (**10**) and (**11**) also protected against Aβ1-42-induced neuronal cell death, with compound (**11**) showing anti-Aβ aggregation effects.

A series of fifteen acetylcholinesterase inhibitors was designed and synthesized by van Greunen, D. G et al. (2019) [102], building on the lead compound 5,6-dimethoxy-1-oxo-2,3-dihydro-1*H*-inden-2-yl 1-benzylpiperidine-4-carboxylate, which exhibited strong inhibitory activity against acetylcholinesterase (IC50 0.03 ± 0.07 μM). Modifications were made to the lead compound, replacing the ester linker with a more metabolically stable amide linker and substituting the indanone moiety with various aryl and aromatic heterocycles. The most potent analog, 1-benzyl-*N*-(1-methyl-3-oxo-2-phenyl-2,3-dihydro-1*H*-pyrazol-4-yl)piperidine-4-carboxamide (**12**), demonstrated IC50 values of 5.94 ± 1.08 μM, respectively, in vitro. Computational predictions suggest that compound (**12**) can cross the blood–brain barrier, and molecular dynamics simulations reveal a strong similarity in binding between compound (**12**) and the FDA-approved acetylcholinesterase inhibitor donepezil.

Series of *N*-benzylpiperidine analogs were synthesized by Sharma, P. (2019) [103] as dual inhibitors of AChE and BACE-1. Compound (**13**) showed the best balanced inhibition. Notably, compound (**13**) had high brain permeability, inhibited AChE-induced Aβ aggregation, and was non-toxic to SH-SY5Y cells. It improved scopolamine-induced cognitive impairment in mice and demonstrated antioxidant and AChE inhibitory properties. Compound (**13**) also showed cognitive improvement in the Morris water maze and good oral absorption.

A novel series of multi-target-directed ligands against AD was developed by combining donepezil and curcumin [104]. Among these, compound (**14**) exhibited strong acetylcholinesterase (AChE) inhibition (IC50 = 187 nM) and the highest selectivity for BuChE over AChE (66.3). Additionally, compound (**14**) inhibited 45.3% of Aβ1-42 self-aggregation at 20 μM and showed significant antioxidant activity. The metal-chelating ability of compound (**14**) was confirmed with a 1:1 stoichiometry for the 14–Cu(II) complex. Moreover, its excellent blood–brain barrier permeability suggests potential efficacy in targeting the central nervous system.

An eco-friendly synthetic route for producing donepezil precursors is presented by Costanzo, P., (2016), [105] utilizing alternative energy sources to enhance yields, regioselectivity, and reaction rates while minimizing waste. The synthesized compounds, which exhibit increased structural rigidity compared to donepezil, were evaluated for AChE inhibition, selectivity against BuChE, side-activity on BACE-1, and effects on SH-SY5Y neuroblastoma cell viability. Two promising lead compounds were identified for a dual therapeutic approach to AD treatment (**15**) and (**16**) (Table 1).

Masitinib (**17**) is a multi-kinase inhibitor (Table 1) that also inhibits fibroblast growth factor receptors and has been identified as a synaptoprotective agent in a dual amyloid precursor protein (APP)/presenilin 1 (PSEN1) mouse model of Alzheimer’s disease (AD) [106,111]. In a Phase 3 clinical trial (NCT01872598) and an ongoing Phase 3 study (NCT05564169), masitinib demonstrated significant cognitive improvements [112]. Additionally, it plays a role in addressing hallmark pathologies of AD, such as tau accumulation, alongside other promising multi-targeted drug candidates aimed at modulating inflammation [112,113].

Dasatinib (**18**), a drug that targets the SRC family tyrosine kinases YES1 and FYN, has been shown to significantly reduce tau phosphorylation in a neuroblastoma cell line overexpressing the mutant tau protein (Table 1). Meanwhile, the transcription factor STAT3 inhibitor C188-9 has demonstrated the ability to alleviate neuroinflammation, tau phosphorylation, and amyloid-beta (Aβ) secretion [114]. Additionally, dasatinib influences the levels of pro-inflammatory and anti-inflammatory cytokines in wild-type mice [108]. Additionally, it plays a role in addressing hallmark pathologies of AD, such as tau accumulation, alongside other promising multi-targeted drug candidates aimed at modulating inflammation [106,113]. A phase I, open-label, proof-of-concept trial was conducted to evaluate the CNS penetrance, safety, feasibility, and efficacy of orally administered senolytic therapy—dasatinib (D) and quercetin (Q)—in early-stage symptomatic Alzheimer’s patients. Findings showed CNS penetrance of dasatinib and supported its safety, tolerability, and feasibility in AD patients. Biomarker data offered mechanistic insights into senolytic effects, warranting confirmation in larger, placebo-controlled studies. ClinicalTrials.gov identifier: NCT04063124.

As indicated in Table 1 and clinical trials, multi-target drugs containing an *N*-benzyl piperazine fragment have shown enhanced efficacy in mitigating cognitive decline and addressing key Alzheimer’s disease pathologies, including amyloid and tau accumulation. The ongoing advancement in our genetic, molecular, and pathological understanding of AD bolsters our optimism that MTDs will significantly transform the treatment landscape for this challenging disease.

Thus, recent studies emphasize the potential of MTDs with *N*-benzylpiperidine or *N*-benzylpiperizine fragments to provide more holistic therapeutic approaches by simultaneously targeting multiple disease mechanisms. This growing body of evidence suggests that these innovative therapies could lead to improved patient outcomes and alter the trajectory of AD management.

In addition, hybrids incorporating donepezil-like pharmacophores, with the *N*-benzylpiperidine moiety as a linker, notably enhance inhibitory activity against both acetylcholinesterase (AChE) and butyrylcholinesterase (BuChE). Furthermore, the addition of donepezil-like pharmacophores not only strengthens monoamine oxidase B (MAO-B) inhibition but also modulates amyloid-beta (Aβ) aggregation and mitigates neurotoxicity.

## 6. Natural Compounds as Multi-Target Drugs for Alzheimer’s Disease Treatment

Natural compounds offer a promising avenue for multi-targeted approaches to Alzheimer’s disease (AD) treatment. By addressing multiple pathological features of AD, these compounds may provide neuroprotection and improve cognitive function. Further research, particularly in optimizing bioavailability and conducting large-scale clinical trials, will be essential in translating these natural compounds into viable therapeutic options for AD.

Natural compounds have emerged as promising multi-target agents for the treatment of AD, given their ability to modulate multiple pathological pathways involved in the disease. These compounds, derived from plants and other natural sources, offer neuroprotective effects by targeting key mechanisms such as amyloid-beta plaque accumulation, tau hyperphosphorylation, oxidative stress, neuroinflammation, and mitochondrial dysfunction—hallmarks of AD pathology.

Below are some key natural compounds being explored as multi-target drugs for AD treatment, some of them included in clinical trials.

### 6.1. Natural Compounds in Clinical Trials for AD Treatment

Two notable examples effective in managing Alzheimer’s symptoms are huperzine A and galantamine, derived from natural sources. Beyond these, many other natural products (NPs) show potential for AD treatment by acting through antioxidant, anti-inflammatory, and neuroprotective mechanisms. While extensive reviews exist on NPs in AD treatment, this section will focus on NPs currently in clinical trials, with key clinical information summarized in Figure 5 [115].

### 6.2. Melatonin

In light of the growing global health crisis posed by AD, Zefan Zhang et al. [116] provide a comprehensive review examining melatonin’s (Figure 6) potential as both a preventive and therapeutic agent. As a naturally occurring hormone with strong antioxidant properties, increasing evidence points to melatonin as a promising candidate in addressing AD-related pathologies. The review highlights several mechanisms, including its possible effects on amyloid-beta accumulation, tau pathology, antioxidant defense, immune response, and circadian rhythm regulation. However, significant gaps remain before clinical application is feasible. These include the need for more randomized clinical trials involving patients with or at risk for AD, determining optimal dosage and timing, and assessing potential side effects, especially with long-term use. The review [116] consolidates current knowledge, identifies these gaps, and proposes future research directions to better understand melatonin’s neuroprotective potential and its role in mitigating AD.

On the positive side, melatonin’s accessibility, affordability, and potential benefits position it as a promising intervention that requires further testing [117]. Studies among both AD dementia populations and preclinical, asymptomatic AD, coupled with biomarker testing, are needed to address remaining gaps for translation [116]. Since peptides, proteins, and hormones can directly reach the brain when administered intranasally via transport and diffusion along the olfactory and trigeminal nerves, future studies should explore whether intranasal melatonin could be a viable therapeutic option for increasing brain melatonin levels in individuals at risk of developing AD. Notably, intranasal melatonin has demonstrated effectiveness in improving sleep in proof-of-concept studies, suggesting that this route of administration could enhance melatonin’s neuroprotective effects and improve patient outcomes [2,116,118]. Further research is essential to evaluate its long-term efficacy, safety, and potential to mitigate AD progression [119].

### 6.3. Cannabidiol

Cannabidiol (CBD) (Figure 7), a non-psychoactive compound from Cannabis sativa, is emerging as a potential therapeutic agent for AD due to its diverse biological effects, including anti-inflammatory, antioxidant, neuroprotective, and anxiolytic properties [120,121]. The research highlights the potential mechanisms by which CBD may mitigate AD-related pathologies [122,123,124]. CBD’s anti-inflammatory effects [125,126] help reduce neuroinflammation by downregulating pro-inflammatory cytokines, potentially slowing disease progression. Its antioxidant properties [127] combat oxidative stress [128], a key contributor to neuronal damage in AD. Preclinical studies suggest CBD also modulates amyloid-beta and tau pathology [129,130], two hallmark features of AD, reducing plaque accumulation and tau hyperphosphorylation. Additionally, CBD promotes neurogenesis [131], potentially enhancing cognitive function by compensating for neuronal loss. Furthermore, CBD’s ability to alleviate anxiety, depression, and sleep disturbances could improve the overall quality of life for AD patients [132]. These multifaceted properties make CBD a promising candidate for AD management, warranting further investigation in clinical trials.

### 6.4. Dronabinol

Dronabinol (Figure 8) is a synthetic delta-9-THC that is indicated in anorexia treatment and loss of weight in HIV patients, nausea, and cancer chemotherapy-related vomiting [133]. A study presented at the International Psychogeriatric Association’s annual meeting (25–27 September 2024 Buenos Aires) found dronabinol to be a safe, effective treatment for agitation in Alzheimer’s disease (Agit-AD) [134]. Led by Dr. Paul Rosenberg from Johns Hopkins, the three-week, placebo-controlled trial with 80 patients showed significant improvements on the Pittsburgh Agitation Scale (PAS) and Neuropsychiatric Inventory (NPI-C) agitation subscales in the dronabinol group. No significant differences in adverse events were noted between groups. Researchers suggest this trial could support “repurposing” dronabinol as a novel treatment for Agit-AD, with promising public health impact.

### 6.5. Curcumin

Curcumin (Figure 9), the active compound in Curcuma longa (turmeric), has gained significant attention as a potential therapeutic agent [135,136] for AD due to its potent anti-inflammatory [137], antioxidant [138], and neuroprotective properties [139,140,141]. Given that AD is characterized by amyloid-beta plaque accumulation, tau tangles, oxidative stress, and chronic neuroinflammation, curcumin’s ability to target these multiple pathological pathways makes it a promising candidate for treatment [135,142,143]. Curcumin has been shown to inhibit amyloid-beta aggregation [144,145], modulate tau phosphorylation [145,146,147], reduce oxidative damage, and suppress neuroinflammation by downregulating NF-κB signaling [148,149]. Additionally, curcumin may promote neurogenesis in the hippocampus [150,151], which could counteract neuronal loss and improve cognitive function. Despite these promising mechanisms, challenges such as curcumin’s low bioavailability, rapid metabolism, and the need for further long-term clinical trials remain. Researchers are developing novel formulations, such as nanoparticles and liposomal curcumin, to enhance its bioavailability. Curcumin’s multi-targeted effects suggest it may be most effective in combination therapies [135]. While curcumin shows great potential for AD treatment, larger clinical studies are essential to confirm its efficacy, safety, and long-term benefits.

### 6.6. Resveratrol

Resveratrol (Figure 10), a natural polyphenol found in grapes, berries, and red wine, has garnered interest as a potential therapeutic agent for AD due to its potent antioxidant and anti-inflammatory properties [152,153,154]. Resveratrol’s ability to target key pathological features of AD, such as amyloid-beta plaque accumulation [155] and tau hyperphosphorylation [156], makes it a promising neuroprotective compound [157]. Resveratrol has been shown to reduce amyloid-beta accumulation [158] by enhancing its clearance, inhibit tau hyperphosphorylation [159], mitigate oxidative stress [160,161] by neutralizing free radicals, and reduce neuroinflammation by downregulating pro-inflammatory cytokines [161,162]. Additionally, resveratrol activates the SIRT1 pathway [163], which is linked to improved amyloid-beta clearance, reduced inflammation [152], and better synaptic plasticity, potentially improving cognitive function. Resveratrol may also protect the integrity of the blood–brain barrier [164,165,166], preventing further exacerbation of AD. Despite its potential, resveratrol’s low bioavailability remains a challenge [167], prompting the exploration of novel delivery systems to enhance its effectiveness. While preclinical studies are promising, further long-term clinical trials are required to establish resveratrol’s efficacy, safety, and optimal dosage in AD patients [168].

### 6.7. Quercetin

Quercetin (Figure 11), a natural flavonoid abundantly found in various fruits and vegetables, has garnered attention as a potential therapeutic agent for AD due to its multifaceted neuroprotective properties [169,170]. Emerging research indicates that quercetin can significantly reduce amyloid-beta levels [171,172], a hallmark of AD pathology, and inhibit tau aggregation [173], thereby addressing two critical aspects of the disease’s progression. Its robust antioxidant activity [174] plays a vital role in protecting neurons from oxidative stress, a key contributor to neuronal damage and cognitive decline in AD. Additionally, quercetin exhibits strong anti-inflammatory effects [175] that help mitigate neuroinflammatory damage, further enhancing its protective capabilities. By targeting multiple pathways [176] involved in AD pathology—such as amyloid-beta accumulation, tau hyperphosphorylation, oxidative stress, and neuroinflammation—quercetin provides a comprehensive approach to neuroprotection. Despite its promising potential [177], challenges related to bioavailability [178] and the need for large-scale clinical trials remain. Future research should focus on optimizing quercetin formulations and exploring its efficacy in combination therapies to establish its role in the prevention and treatment of AD [179]. In addition, quercetin and dasatinib are being explored as potential therapies for early-stage AD due to their senolytic properties, which may help clear senescent cells that contribute to neurodegeneration. ClinicalTrials.gov: NCT04063124.

### 6.8. Licochalcone A

Licochalcone A (LCA) (Figure 12) is a natural compound derived from the root of *Glycyrrhiza inflata* (licorice). It has garnered attention for its potential therapeutic effects in AD with strong antioxidant properties, helping to reduce oxidative stress, a significant factor in AD pathology [180]. LCA may protect neurons from damage caused by amyloid-beta (Aβ) plaques and tau protein hyperphosphorylation, both hallmarks of AD [181]. It has been shown to inhibit inflammatory pathways, which could mitigate neuroinflammation associated with AD [182]. Some studies suggest that LCA may enhance cholinergic function, potentially improving cognitive deficits [183].

### 6.9. Pinitol

Pinitol (Figure 13), a naturally occurring cyclitol present in various plants, particularly in pine nuts and soy, has been studied for its potential therapeutic benefits in Alzheimer’s disease. Here is a summary of its properties and role in clinical research: Pinitol has insulin-mimicking effects that may boost glucose metabolism in the brain, an important factor given the link between insulin resistance and AD pathology. Additionally, it may provide neuroprotection by mitigating oxidative stress and inflammation, both of which significantly contribute to neuronal damage in AD. Pinitol may also enhance cholinergic signaling, potentially supporting cognitive function [184].

The discussed natural compounds—melatonin, quercetin, pinitol, resveratrol, cannabidiol, licochalcone A, curcumin, and dronabinol—exhibit unique structural and functional properties that may contribute to their therapeutic effects in Alzheimer’s disease. Many of them, such as quercetin and curcumin, contain multiple hydroxyl (–OH) groups, crucial for their antioxidant activity, although they are all antioxidants. Compounds like quercetin and resveratrol feature aromatic rings that enhance their free radical scavenging ability, while chiral centers in cannabidiol and dronabinol may influence their pharmacological effects. These compounds possess strong antioxidant and anti-inflammatory properties, helping to neutralize oxidative stress and reduce neuroinflammation associated with AD. Melatonin and cannabidiol can also protect against Aβ-induced neurotoxicity, while pinitol and quercetin may enhance cholinergic signaling, potentially improving cognitive function. Some, like quercetin, also have senolytic properties, aiding in the elimination of senescent cells. The combination of their features positions these compounds as promising candidates for AD treatment, targeting multiple pathways involved in the disease. Further research and clinical trials are needed to establish their efficacy and safety in managing Alzheimer’s disease.

## 7. Conclusions

Recent research focuses on developing novel bioactive hybrid compounds that target multiple pathways concurrently. Polypharmacology, which involves drugs acting on multiple targets, has the potential to reduce toxicity and drug interactions compared to traditional single-target therapies. Hybrid compounds that combine multiple bioactive elements can offer improved efficacy and cost-effective solutions. Although approved treatments are limited, heterocyclic compounds based on *N*-benzylpiperidine fragments have shown promise in AD drug discovery. This review highlights the importance of the *N*-benzylpiperidine structure as part of multi-targeted drugs (MTDs) and its unique properties that contribute to its therapeutic potential, especially for Alzheimer’s disease (AD). These properties make *N*-benzylpiperidine derivatives promising candidates in drug discovery, particularly for multi-targeted approaches in treating Alzheimer’s disease targeting oxidative stress, cholinergic deficits, neuroinflammation, amyloid-beta (Aβ) accumulation, and tau protein hyperphosphorylation in AD. This review also shows the potential of various natural compounds with unique structural and functional properties as multi-targeted therapies for Alzheimer’s disease (AD), addressing multiple disease pathways. Compounds such as melatonin, quercetin, pinitol, resveratrol, cannabidiol (CBD), licochalcone A, curcumin, and dronabinol have demonstrated therapeutic effects in clinical studies, including improvements in cognitive function, reductions in inflammation, and neuroprotective properties. By simultaneously targeting oxidative stress, inflammation, and neuroprotection, these compounds present a promising approach to AD treatment. However, further clinical trials are essential to validate their efficacy and safety profiles in the context of Alzheimer’s disease management. All these summarized findings suggest that combining *N*-benzylpiperidine or *N*-benzylpiperazine fragments with natural products could lead to the creation of hybrid molecules with enhanced pharmacokinetic and pharmacodynamic properties, minimized side effects, and improved therapeutic efficacy in targeting complex diseases like Alzheimer’s. This approach may support the development of multi-targeted drugs that address various pathways in neurodegeneration, ultimately contributing to more effective and safer treatment options for AD.

## Figures and Tables

**Figure 1 molecules-29-05314-f001:**
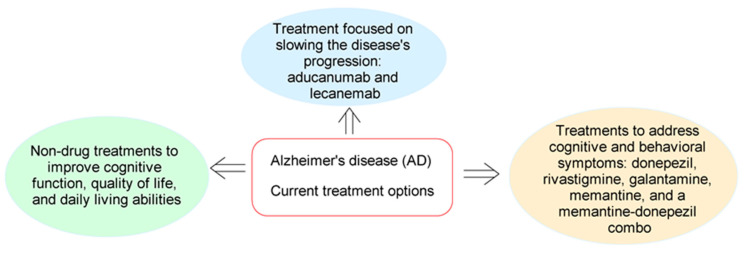
Current treatment options for Alzheimer’s disease.

**Figure 2 molecules-29-05314-f002:**
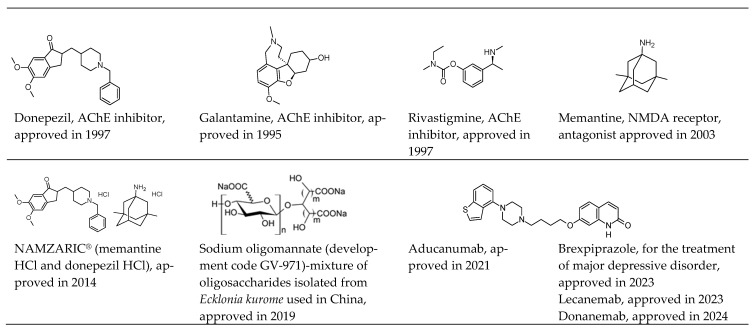
Current anti-AD drugs approved by FDA/China. Aducanumab, lecanemab, and donanemab are monoclonal antibodies that target amyloid-beta plaques.

**Figure 3 molecules-29-05314-f003:**
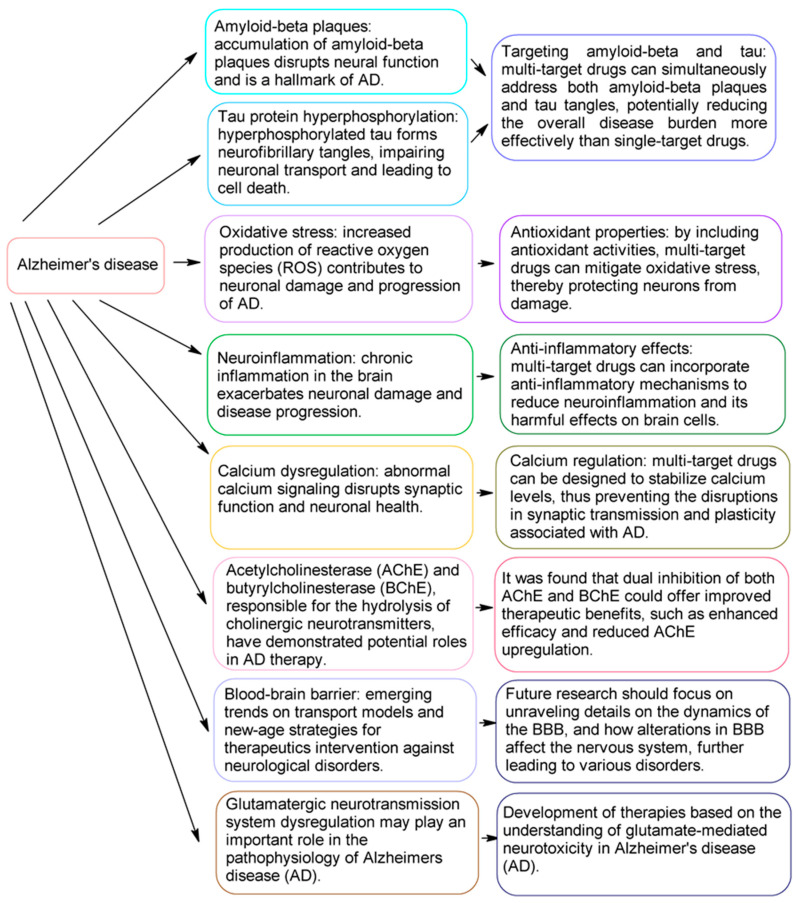
Hybrid multi-target therapeutic compounds addressing the multifaceted nature of Alzheimer’s disease (AD).

**Figure 4 molecules-29-05314-f004:**
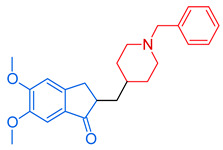
Donepezil possesses a chemical structure characterized by a bicyclic system that includes an indanone (or indole-like) ring connected to the *N*-benzylpiperidine ring, playing a crucial role in its pharmacological activity.

**Figure 5 molecules-29-05314-f005:**
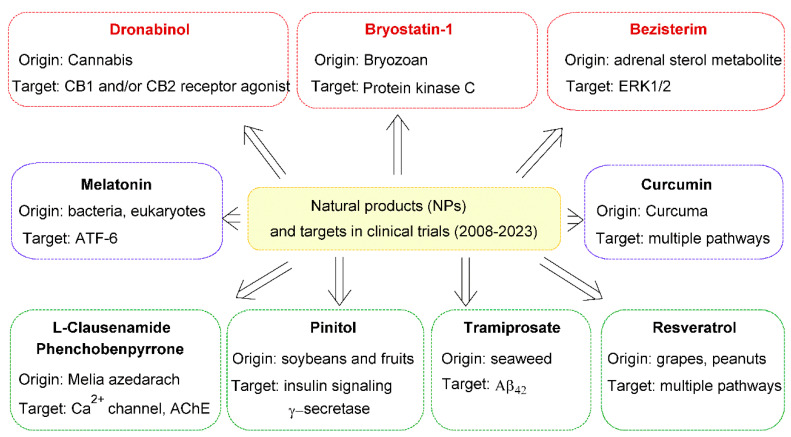
Natural compounds in clinical trials for AD treatment, 2008–2023 [115].

**Figure 6 molecules-29-05314-f006:**
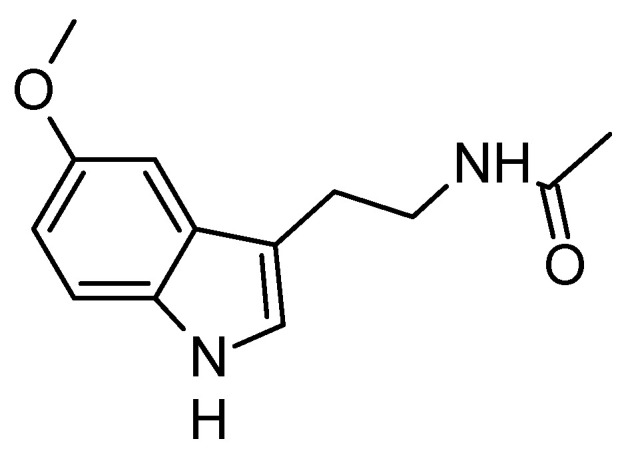
Melatonin (*N*-acetyl-5-methoxytryptamine), a tryptophan metabolite synthesized mainly in the pineal gland.

**Figure 7 molecules-29-05314-f007:**
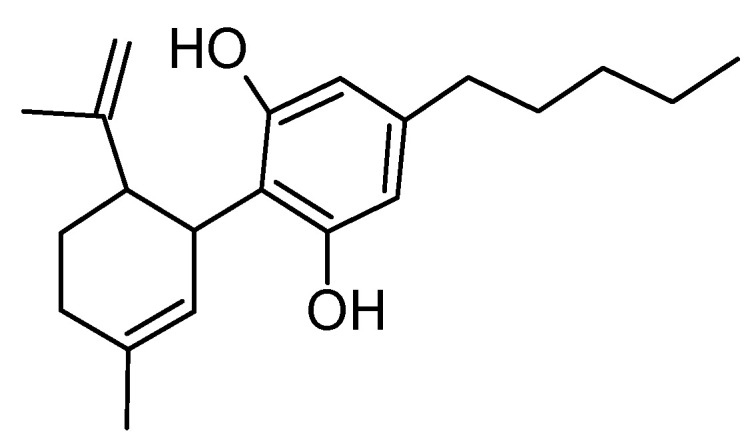
Structure of CBD. This compound mainly contains a cyclohexene ring, a phenolic ring, and a pentyl side chain in the structure.

**Figure 8 molecules-29-05314-f008:**
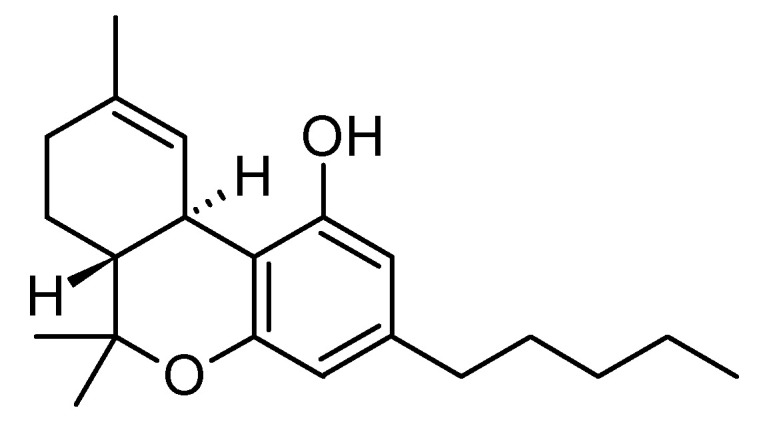
Dronabinol, the synthetic form of delta-9-tetrahydrocannabinol (THC), is a psychoactive compound and primary active component found in cannabis.

**Figure 9 molecules-29-05314-f009:**
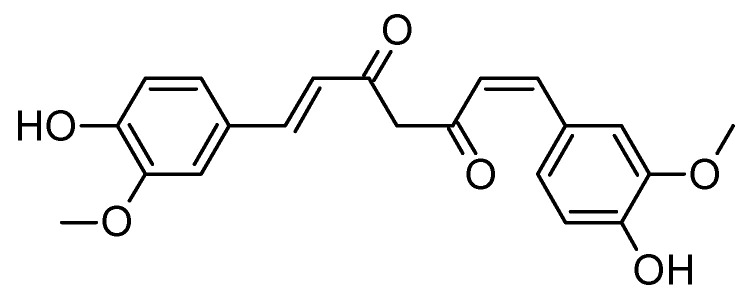
The structure of curcumin with the chemical name of diferuloylmethane.

**Figure 10 molecules-29-05314-f010:**
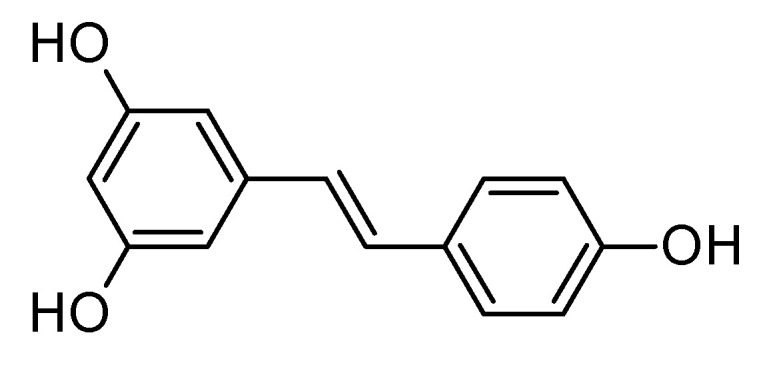
Structure of resveratrol, a natural polyphenolic compound.

**Figure 11 molecules-29-05314-f011:**
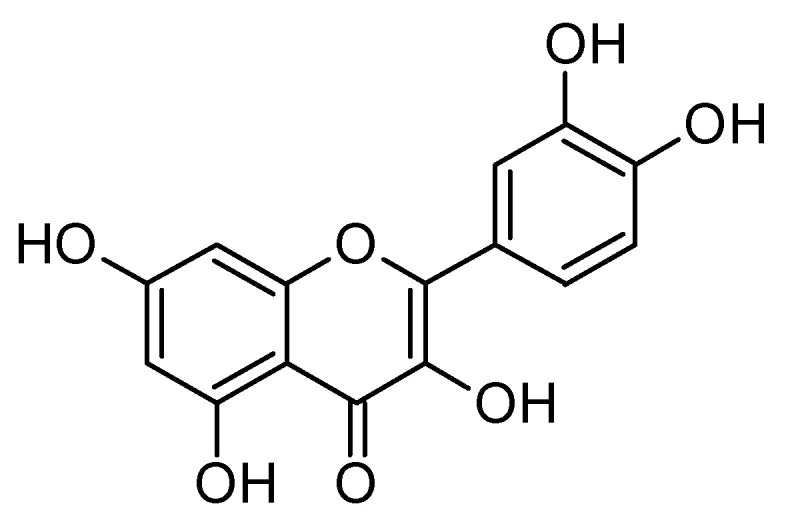
Chemical structure of quercetin (3,3′,4′,5,7-pentahydroxyflavone). Quercetin has a polyphenolic structure characterized by two benzene rings (A and B) connected by a three-carbon chain that includes a ketone group.

**Figure 12 molecules-29-05314-f012:**
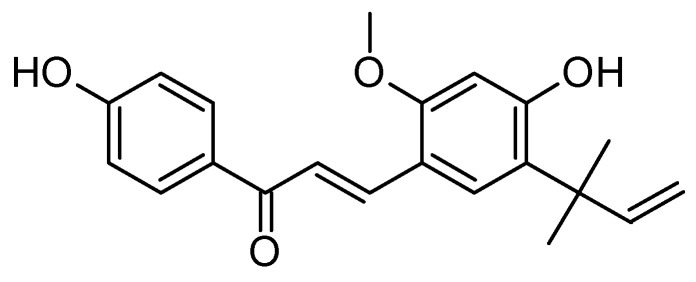
Licochalcone A is a flavonoid compound derived from the roots of Glycyrrhiza uralensis (licorice) and has a unique chalcone structure, which differentiates it from other flavonoids.

**Figure 13 molecules-29-05314-f013:**
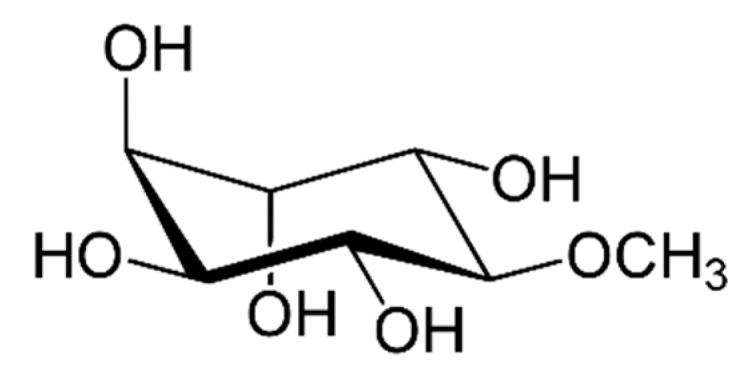
Pinitol, a naturally occurring sugar alcohol, is a derivative of inositol, specifically known as 3-O-methyl-D-chiro-inositol.

**Table 1 molecules-29-05314-t001:** Novel donepezil-based hybrids—activities (NT—not tested).

Hybrid Compound	AChE Inhibitor, IC50 μM BchE Inhibitor, IC50 μM	β-Amyloid Antiaggregation	Antioxidant Potential	BBB Permeability	Other Activities	Experimental Studies	References
Indole-piperidine amides 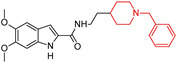 (**1**)	-EeAChE: IC50 0.52 μM -eqBChE: IC50 19.16 μM -hAChE: IC50 0.32 μM -hBChE: IC50 0.39 μM	NT	NT	yes	NT	in vitro	[90]
Indanone/benzofuranone and piperidine hybrids 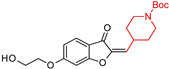 (**2**)	NT	NT	NT	yes	-Good neuroprotection; -Low cytotoxicity.	in vitro and in vivo in rats	[91]
Hybrid structures of baicalein and donepezil 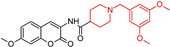 (**3**)	AChE: IC50 0.05 ± 0.02 µM BuChE: IC50 0.946 µM	yes	yes	yes	-Protect nerve cells.	in vitro	[92]
Benzamide derivatives 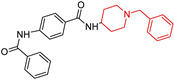 (**4**)	AChE: IC50 0.14 ± 0.03 nM	NT	NT	NT	NT	in vitro	[93]
Benzylpiperazinequinoline hybrids 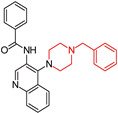 (**5**)	eqBChE: IC50 0.059 ± 0.006 μM, hBChE: IC50 0.162 ± 0.069 μM AChE: NA (Not Active) SI: eeAChE/eqBChE: 508.47 eeAChE/hBChE: 190.44	yes	yes	yes	-Metabolic stability; -High oral bioavailability; -Protected neural cells from toxicity and inflammation in vitro; -Weak toxicity in neural cells (SH-SY5Y, anti-neuroinflammatory effect); -Improving cognitive function in mouse models.	in vitro and in vivo in rats	[94]
Indole- and/or donepezil-like hybrids 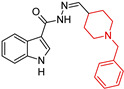 (**6a**) 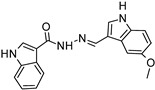 (**6b**)	6a: AChE: IC50 (10.76 ± 1.66 μM) 26.32 ± 3.11 hBChE: IC50 26.32 ± 3.11 SI:AchE = 2.45 6b:BChE IC50 21.12 ± 1.48 μM; SI: BChE = 47.34	6a yes 6b yes	6a yes 6b yes	6a yes 6b yes	-Effectively targets AD biomarkers Aβ1-42 and pTAU in a rat model; -Facilitates non-amyloidogenic signaling through MT1A and MT2B/ERK/CREB pathways.	in vitro, in vivo and ex vivo	[95,96,97]
Piperazine and *N*-benzylpiperidine hybrids of 5-phenyl-1, 3, 4-oxadiazol-2-thiol 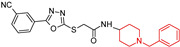 (**7a**) 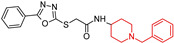 (**7b**)	7a: hAChE: IC50 0.076 μM hBChE: IC50 1.204 μM hBChE-1: IC50 0.230 μM 7b: hAChE: IC50 0.113 μM hBChE: IC50 1.480 μM hBChE-1: IC50 0.318 μM	7a yes 7b yes	NT	7a yes 7b yes	-β-secretase-1 (hBACE-1); -Improved learning and memory; -Reduced MDA, NO levels; -Increased GSH; -Lowered pro-inflammatory cytokines.	in vitro, in vivo, and ex vivo	[98]
*N*-Benzyl piperidine derivatives 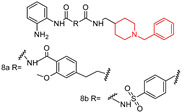 (**8a**) and (**8b**)	8a: HDAC: IC50 0.17 μM, AchE: IC50 6.89 μM; histone deacetylases (HDACs) 8b: HDAC: IC50 0.45 μM, AchE: IC50 3.22 μM).	yes	yes	NT	-Neuroprotective effects in PC-12 cells; -Good selectivity for AchE; -Protected PC-12 cells from H_2_O_2_ induced cytotoxicity.	in vitro	[99]
Pyrazolopyridine and tetrahydroacridine (THA) hybrids 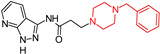 (**9**)	hAChE and binding to the peripheral anionic site (PAS)	yes	yes	yes	-Safety in hepG2 cells LD50 values; -Exceeding 120 mg/kg.	in vitro and in vivo	[100]
*N*-alkylpiperidine carbamates 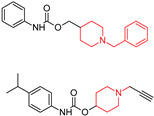 (**10**) and (**11**)	10: multiple AchE: IC50 = 7.31 μM, BchE: (IC50 = 0.56 μM) and MAO-B: (IC50 = 26.1 μM) 11: selective MAO-B: (IC50 = 0.18 μM).	yes	NT	yes	-Not cytotoxic to human neuronal-like SH-SY5Y; -Liver HepG2 cells; -Inhibit monoamine oxidases [monoamine oxidase A (MAO-A and monoamine oxidase B (MAO-B)].	in vitro	[101]
*N*-benzylpiperidine carboxamide derivatives 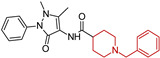 (**12**)	AchE: IC50: 5.94 ± 1.08 μM	NT	NT	yes	NT	in vitro	[102]
*N*-benzylpiperidine analogs 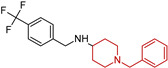 (**13**)	AchE: (IC50: 1. 0.11 ± 0.02) BchE: (IC50 = 3.0 ± 0.06) hBACE-1: (IC50 = 0.22 ± 0.02) hAChE SI = 28.2	yes	yes	yes	-Devoid of neurotoxicity towards SH-SY5Y neuroblastoma cell lines; -Amelioration of scopolamine- and Aβ-induced cognitive impairment in AD rat models.	in vitro and in vivo	[103]
Donepezil and curcumin hybrids 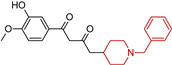 (**14**)	AchE: IC50 = 187 nM highest selectivity for BuChE over AChE (66.3)	yes	yes	yes	NT	in vitro and in silico	[104]
Donepezil analogs 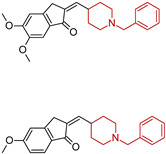 (**15**) and (**16**)	hAChE: (IC50 = 0.058 ± 0.033) BuChE: (IC50 = 4.740 ± 0.750) hAChE: (IC50 = 0.043 ± 0.007 BuChE: (IC50 = 5.734 ± 0.130	NT	NT	NT	-Did not influence the cell viability in SH-SY5Y neuroblastoma cells.	in vitro	[105]
Masitinib Clinical trial—Phase 3 study is ongoing. NCT01872598, NCT05564169 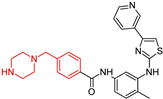 (**17**)	no	yes	no	yes	-Multi-kinase inhibitor with additional FGF receptor inhibition; characterized as synaptoprotective agent—tau protein signaling pathway; -Prevention of synaptic damage; -Significantly improved cognition in Phase 3 study.	in vitro and in vivo	[106,107]
Dasatinib plus quercetin Clinical trial—Phase 1/2 study NCT04063124, NCT04785300, 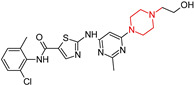 (**18**)	no	yes	yes	yes	Senolytic for ephrins, PI3Kδ, p21, BCL-xL, and plasminogen-activator inhibitor 2	in vitro and in vivo	[108,109,110]

## Data Availability

The original data presented in the study are openly available in the reference/accession number.

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
