# Peer review of "New Insights into the Development of Donepezil-Based Hybrid and Natural Molecules as Multi-Target Drug Agents for Alzheimer’s Disease Treatment"

_molecules, 2024, doi:10.3390/molecules29225314_

Round 1

Reviewer 1 Report

Comments and Suggestions for Authors

The manuscript „Recent advancements in the development of multi-target drugs for Alzheimer`s disease treatmant“ addresses a relevant topic and it could make a relevant contribution to the field. However, there is several things that require the author’s attention. The most important thing is that the mayority of the references used for this review are already published review articles about this theme. This article should give some new insights about this theme, because in this form its repetition of already published articles. I strongly advise and encourage authors to take time and look at research articles about new compounds designed, synthesized and evaluated as potential multi-target new drugs for Alzheimer`s diesease and to supplement their article with new knowledge, prospectives and insights and send it back. . Also, general comments are: there is too much repetition of already mentioned facts. The authors repeatedly describe the pathophysiology of Alzheimer's disease, its multifactoriality, and repeatedly list and describe the drugs used in its treatment throughout the entire text.

 In the sections no. 1 and 2 there is no marked reference in the text by which the authors confirm their claims.

-          In the section Introduction line no. 43  is repetition of line no. 41 

-          In the same section in the line no. 44 authors mentioned only two drugs that are targeting beta-amyloid plaques, but there is another one Donanemab (Kisunla), which is approved in July 2024

-          Also in the Introduction, from line no. 40 to line 65, there is no reference by which the authors confirm their claims

-          In the section Current drugs for treatments in the line no. 68 is said:  As of now, eight drugs are commonly used to treat Alzheimer's disease (Figure 1)….but authors mentioned only seven: aducanumab, lecanemab, donepezil, rivastigmine, galantamine, memantine, and a memantine-donepezil combination (Namzaric)…the Donanemab should be mentioned

-          In the section Treatments focused on slowing Alzheimer’s progression: aducanumab and lecanemab, there is no reference by which the authors confirm their claims. This is a review article and there should be the list of referenes from which the authors took their claims.

Moreover, authors didn`t mention the third drug that target beta-amyloid plaques in the brain, Donanemab (Kisunla), which is approved in July 2024

-          The reference are also missing from the section Treatments to address cognitive and behavioral symptoms – donepezil, rivastigmine, galantamine, memantine, and a memantine-donepezil combination. In the line no. 107 it is said: that enhance neurotransmitter levels, which neurotransmitter? (Acetylcholine)

In section 3. Advants of multy-target drugs in Alzheimer's Table 1. the is unreadable, claims are repetitive and incomprehensible. I suggest to author to incorporate the statements presented in Table 1 into the text of this chapter and to pay attention to the repetitions that are already in the text of this section

4. Molecular pathogenesis of AD and hybrid multitarget therapeutic compounds addressing the multifaceted nature of AD

- Line no. 187-188 which hydrolyze cholinergic neurotransmitters it`s untrue…cholinesterases are responsabile for hydrolisis of neurotransmitter acetylcholine, which decreased levels are one of the phatological features of AD.

- Genreal comment, this section is repetition of previous sections.  It should start from the line no.231 because the things said in lines 179 to 230 are already mentioned in previous section and as such represent an unnecessary repetition.

5. Novel Donepezil-based hybrids with a focus on N-benzylpiperidine derivatives for 242 targeting AD (2014-2024)

-          Lines no 244-250 Multi-target-directed ligands (MTDLs)….are the repetition of already listed definitions and as such is unnecessary. Suggestion is to start this part with the story about donepezil analogues…without repetiting alreday explained definitions

-          Table 2. Novel donepezil-based hybrids is unreadable in this form, and repetitve. I suggest to author to put only the structures of the compounds and their targetets in the Table, because the explanation of every compound  is already in the text of the section.

-          Line no 307 is repetition…

-          The structure of compounds  3b, d5 and d10 are missing from the Table 2

6. Natural compounds as multitarget drugs for Alzheimer's disease treatment

- once again, there is only one reference at the begining of this section

Conclusion

 Emphasize the contributions your literature review makes to the field. Discuss how your review  sheds new light on existing knowledge.

 Reference

There is several references that are listed more than once.

Author Response

Response to Reviewer 1 Comments

Thank you very much for taking the time to review this manuscript. Please find the detailed responses below and the corresponding corrections highlighted in track changes in the re-submitted file.

Comments and Suggestions for Authors

Reviewer 1: The manuscript „Recent advancements in the development of multi-target drugs for Alzheimer`s disease treatmant“ addresses a relevant topic and it could make a relevant contribution to the field. However, there is several things that require the author’s attention. The most important thing is that the mayority of the references used for this review are already published review articles about this theme. This article should give some new insights about this theme, because in this form its repetition of already published articles. I strongly advise and encourage authors to take time and look at research articles about new compounds designed, synthesized and evaluated as potential multi-target new drugs for Alzheimer`s diesease and to supplement their article with new knowledge, prospectives and insights and send it back.

Response and Revisions: Thank you very much for your valuable comments.

Yes, there are numerous articles discussing Alzheimer's disease and the benefits of multi-targeted drugs, which we aimed to summarize. In this article, we highlight how multi-targeted approaches can effectively address the complex nature of Alzheimer's pathology by targeting multiple mechanisms, such as amyloid-beta accumulation, tau phosphorylation, and neuroinflammation. We provide recent examples involving natural products, some of which are currently in clinical trials, alongside donepezil-based hybrids. These examples align with our current project's objectives and research activities, providing valuable insights to guide further research and therapeutic development. They highlight the significance of the N-benzylpiperidine fragment and its modifications in the structure of the tested compounds, some of which are currently in clinical trials (see Table 1). Following your recommendations, we conducted a thorough literature review and focused specifically on multi-targeted drugs in clinical trials that contain a modified N-benzylpiperidine fragment. Given the vast chemical diversity in this field, we opted to limit our discussion to these specific compounds to maintain clarity and relevance in this review.

Based on the new insights in the field, we have revised our conclusion as follows:

„Recent research focuses on developing novel bioactive hybrid compounds that target multiple pathways concurrently. Polypharmacology, which involves drugs acting on multiple targets, has the potential to reduce toxicity and drug interactions compared to traditional single-target therapies. Hybrid compounds that combine multiple bioactive elements can offer improved efficacy and cost-effective solutions Although approved treatments are limited, heterocyclic compounds based on N-benzylpiperidine fragments have shown promise in AD drug discovery. This review highlights the importance of the N-benzylpiperidine structure as part of multi-targeted drugs (MTDs) and its unique properties that contribute to its therapeutic potential, especially for Alzheimer's disease (AD). These properties make N-benzylpiperidine derivatives promising candidates in drug discovery, particularly for multi-targeted approaches in treating Alzheimer's disease targeting oxidative stress, cholinergic deficits,  neuroinflammation,  amyloid-beta (Aβ) accumulation, and tau protein hyperphosphorylation in AD. This review also shows the potential of various natural compounds whit unique structural and functional properties as multi-targeted therapies for Alzheimer's disease (AD), addressing multiple disease pathways. Compounds such as melatonin, quercetin, pinitol, resveratrol, cannabidiol (CBD), licochalcone A, curcumin, and dronabinol have demonstrated therapeutic effects in clinical studies, including improvements in cognitive function, reduction of inflammation, and neuroprotective properties. By simultaneously targeting oxidative stress, inflammation, and neuroprotection, these compounds present a promising approach to AD treatment. However, further clinical trials are essential to validate their efficacy and safety profiles in the context of Alzheimer's disease management. All these summarized findings suggest that combining N-benzylpiperidine fragments with natural products could lead to the creation of hybrid molecules with enhanced pharmacokinetic and pharmacodynamic properties, minimized side effects, and improved therapeutic efficacy in targeting complex diseases like Alzheimer's. This approach may support the development of multitargeted drugs that address various pathways in neurodegeneration, ultimately contributing to more effective and safer treatment options for AD.“

According to your suggestion, "I strongly advise and encourage authors to take time to review research articles on new compounds designed, synthesized, and evaluated as potential multi-target drugs for Alzheimer's disease," we have also added text and included two new hybrid multi-target compounds in clinical trials in Table 2.

Masitinib is a multi-kinase inhibitor that also inhibits fibroblast growth factor receptors and has been identified as a synaptoprotective agent in a dual amyloid precursor protein (APP)/presenilin 1 (PSEN1) mouse model of Alzheimer’s disease. In a Phase 3 clinical trial (NCT01872598) and an ongoing Phase 3 study (NCT05564169), Masitinib demonstrated significant cognitive improvements. Additionally, it plays a role in addressing hallmark pathologies of AD, such as tau accumulation, alongside other promising multi-targeted drug candidates aimed at modulating inflammation.

Dasatinib, a drug that targets the SRC family tyrosine kinases YES1 and FYN, has been shown to significantly reduce tau phosphorylation in a neuroblastoma cell line overexpressing the mutant tau441 protein. Meanwhile, the transcription factor STAT3 inhibitor C188-9 has demonstrated the ability to alleviate neuroinflammation, tau phosphorylation, and amyloid-beta (Aβ) secretion. Additionally, Dasatinib influences the levels of pro-inflammatory and anti-inflammatory cytokines in wild-type mice. Additionally, it plays a role in addressing hallmark pathologies of AD, such as tau accumulation, alongside other promising multi-targeted drug candidates aimed at modulating inflammation. A phase I, open-label, proof-of-concept trial was conducted to evaluate the CNS penetrance, safety, feasibility, and efficacy of orally administered senolytic therapy—dasatinib (D) and quercetin (Q)—in early-stage symptomatic Alzheimer’s patients. Findings showed CNS penetrance of D and supported its safety, tolerability, and feasibility in AD patients. Biomarker data offered mechanistic insights into senolytic effects, warranting confirmation in larger, placebo-controlled studies. ClinicalTrials.gov identifier: NCT04063124.

As indicated in the Table 2 and clinical trials, multi-target drugs containing a N-benzyl piperazine fragment have shown enhanced efficacy in mitigating cognitive decline and addressing key Alzheimer’s disease (AD) pathologies, including amyloid and tau accumulation. The ongoing advancement in our genetic, molecular, and pathological understanding of AD bolsters our optimism that MTDs will significantly transform the treatment landscape for this challenging disease.

In conclusion, hybrids incorporating donepezil-like pharmacophores, with the N-benzylpiperidine moiety as a linker, notably enhance inhibitory activity against both acetylcholinesterase (AChE) and butyrylcholinesterase (BuChE). Furthermore, the addition of donepezil-like pharmacophores not only strengthens monoamine oxidase B (MAO-B) inhibition but also modulates amyloid-beta (Aβ) aggregation and mitigates neurotoxicity.

Reviewer 1: Also, general comments are: there is too much repetition of already mentioned facts. The authors repeatedly describe the pathophysiology of Alzheimer's disease, its multifactoriality, and repeatedly list and describe the drugs used in its treatment throughout the entire text.

Response and Revisions: We have revised the text and minimized repetition onto the pathophysiology of Alzheimer's disease as follows:

The text: „The pathogenesis of Alzheimer’s disease (AD) is primarily marked by the buildup of two key proteins: extracellular amyloid-beta (Aβ) plaques and neurofibrillary tangles (NFTs) composed of hyperphosphorylated tau protein (Fig. 2). Additionally, factors such as synapse loss, oxidative stress, and neuronal death are also implicated and often occur alongside these primary markers. In the late 1970s, research initially concentrated on two enzymes - acetylcholinesterase (AChE) and butyrylcholinesterase (BChE) - which hydrolyze cholinergic neurotransmitters. Studies highlighted their potential roles in AD treatment It was found that dual inhibition of both AChE and BChE could offer improved therapeutic benefits, such as enhanced efficacy and reduced AChE upregulation. This led researchers to focus on developing inhibitors that target both enzymes  Another key aspect of AD pathology is the amyloid cascade, involving the accumulation of Aβ peptide (Fig. 2) in the brain. Abnormal processing of amyloid precursor protein (APP) by β-secretase and γ-secretase enzymes results in the formation of Aβ40 and Aβ42 monomers, which then aggregate into insoluble fibrils and plaques. These plaques disrupt proteasome function, alter intracellular calcium (Ca2+) levels, and impair mitochondrial function, leading to increased neurotoxicity and fibril formation. Additionally, tau protein, which normally stabilizes microtubules, becomes hyperphosphorylated in AD (Fig. 1), causing disruptions in axonal transport and contributing to neuronal death through the formation of NFTs (76-78). The central nervous system (CNS) is protected by the blood-brain barrier (BBB) and cerebrospinal fluid (CSF) barrier (79, 80). The BBB features uptake and efflux transporters that manage the movement of solutes into and out of the brain. It plays a key role in Alzheimer's disease (AD) by regulating amyloid-beta (Aβ) transport and aggregation within brain tissue, making it crucial for improving drug delivery to the brain (81-83). Oxidative stress is a significant factor in AD-related neuronal cell death (63, 84-87). Monoamine oxidase (MAO) disrupts neurotransmitter metabolism and contributes to Aβ neurotoxicity by increasing reactive oxygen species (ROS) production, which leads to mitochondrial stress (84, 88, 89). The “neuroinflammatory hypothesis” posits that excessive inflammatory responses in the brain can worsen damage and accelerate neurodegeneration (89-92). Calcium (Ca2+) signaling, vital for neurotransmitter release and neuronal function, can become detrimental when levels are persistently high, triggering ROS production and disrupting synaptic plasticity (93-96). In AD, MAO-B accumulation near plaques exacerbates oxidative stress and neuronal damage (88, 89, 97, 98). Current anti-Alzheimer’s drugs function through various mechanisms. For example, donepezil is a noncompetitive cholinesterase inhibitor, while rivastigmine inhibits both acetyl- and butyrylcholinesterases in a pseudo-irreversible way. Galantamine not only inhibits acetylcholinesterase but also enhances nicotinic acetylcholine receptor activity, and memantine regulates glutamatergic transmission. Although combination therapies have been explored to address cognitive decline, only the combination of cholinesterase inhibitors and memantine has shown expected success in clinical trials. Despite promising preclinical results, many other combinations have failed in clinical settings. While cholinesterase inhibitors are generally safe, they can cause significant side effects, especially at higher doses. Furthermore, conventional drugs often have low bioavailability and chemical instability, leading to additional side effects. For instance, high doses of rivastigmine may cause adverse effects, while short half-lives of drugs like tacrine, galantamine, and rivastigmine present dosing challenges. Heterocyclic compounds, including cholinesterase inhibitors and NMDA receptor antagonists, show potential for AD treatment but have limitations.“

was shorted as: „As mentioned above the pathogenesis of Alzheimer’s disease (AD) involves amyloid-beta (Aβ) plaques, neurofibrillary tangles (NFTs), synapse loss, oxidative stress, and neuronal death. Initial research focused on acetylcholinesterase (AChE) and butyrylcholinesterase (BChE) enzymes, targeting dual inhibition for potential therapeutic benefits (63-65). Additionally, factors such as synapse loss, oxidative stress, and neuronal death are also implicated and often occur alongside these primary markers (66-69). The amyloid cascade involves Aβ peptide aggregation, disrupting cellular functions and increasing neurotoxicity (70-75). Other factors include the blood-brain barrier’s role(76-78) in drug delivery, oxidative stress (63, 79-82), neuroinflammation(83-86), and calcium signaling disruptions (87-89).“

Reviewer 1: In the sections no. 1 and 2 there is no marked reference in the text by which the authors confirm their claims.

Response and Revisions: we added references in the text by which we confirm ouer claims, mentioned with red.

-          In the section Introduction line no. 43  is repetition of line no. 41 

Response and Revisions: Thank you for your note. We have revised the two sentences as follows:

“Currently, 156 clinical trials are exploring brain changes, such as tau protein accumulation and inflammation, as potential therapeutic targets for Alzheimer's disease.“

Reviewer 1: -          In the same section in the line no. 44 authors mentioned only two drugs that are targeting beta-amyloid plaques, but there is another one Donanemab (Kisunla), which is approved in July 2024

Response and Revisions: Thank you very much for the valuable advice. We have added the latest drug and revised paragraph 44 as follows:

„Current treatment options, such as Aducanumab, Lecanemab and Donanemab, which target beta-amyloid plaques, represent significant advances but come with limitations. Aducanumab is being discontinued, while Lecanemab and Donanemab have shown moderate benefits in the early stages of Alzheimer's, which includes mild cognitive impairment (MCI) or mild dementia stage of disease.“

Reviewer 1: -          Also in the Introduction, from line no. 40 to line 65, there is no reference by which the authors confirm their claims

Response and Revisions: We apologize for the oversight, but we inadvertently did not submit the version with the finalized formatting of the references. In the current version, references have been added both in the introduction, from paragraph 40 to 65, and duplicate citations of authors have been avoided.

Reviewer 1: -          In the section Current drugs for treatments in the line no. 68 is said:  As of now, eight drugs are commonly used to treat Alzheimer's disease (Figure 1)….but authors mentioned only seven: aducanumab, lecanemab, donepezil, rivastigmine, galantamine, memantine, and a memantine-donepezil combination (Namzaric)…the Donanemab should be mentioned

Response and Revisions Donanemab (Kisunla) has been added in the paragraf "2.1. Treatments focused on slowing Alzheimer’s progression: aducanumab, lecanemab and donanemab" and in the Figure 2., and the citations of authors have been added.

Reviewer 1: -          In the section Treatments focused on slowing Alzheimer’s progression: aducanumab and lecanemab, there is no reference by which the authors confirm their claims. This is a review article and there should be the list of referenes from which the authors took their claims.

Response and Revisions: Donanemab has been added in the paragraf "2.1. Treatments focused on slowing Alzheimer’s progression: aducanumab, lecanemab and donanemab" and in the Figure 2.

We have added references.

Reviewer 1: Moreover, authors didn`t mention the third drug that target beta-amyloid plaques in the brain, Donanemab (Kisunla), which is approved in July 2024

Response and Revisions: Donanemab (Kisunla) has been added.

Reviewer 1: -          The reference are also missing from the section Treatments to address cognitive and behavioral symptoms – donepezil, rivastigmine, galantamine, memantine, and a memantine-donepezil combination. 

Response and Revisions: We have added references

Reviewer 1: In the line no. 107 it is said: that enhance neurotransmitter levels, which neurotransmitter? (Acetylcholine)

Response and Revisions: In line 107, we have revised the sentence as follows: Donepezil, rivastigmine, and galantamine are cholinesterase inhibitors that enhance acetylcholine levels, while memantine is an NMDA receptor antagonist that regulates glutamate activity.

Reviewer 1: In section 3. Advants of multy-target drugs in Alzheimer's Table 1. the is unreadable, claims are repetitive and incomprehensible. I suggest to author to incorporate the statements presented in Table 1 into the text of this chapter and to pay attention to the repetitions that are already in the text of this section

Response and Revisions: Following your recommendation, Table 2 has been removed and converted to red colored text.

Reviewer 1: 4. Molecular pathogenesis of AD and hybrid multitarget therapeutic compounds addressing the multifaceted nature of AD

- Line no. 187-188 which hydrolyze cholinergic neurotransmitters it`s untrue…cholinesterases are responsabile for hydrolisis of neurotransmitter acetylcholine, which decreased levels are one of the phatological features of AD.

Response and Revisions: the sentence has been removed

Reviewer 1: - Genreal comment, this section is repetition of previous sections.  It should start from the line no.231 because the things said in lines 179 to 230 are already mentioned in previous section and as such represent an unnecessary repetition.

 Response and Revisions: the text has been shortened and revised as follows:

“Given the multifactorial nature of AD, single-target treatments have shown limited success, underscoring the need for hybrid multitarget compounds. These hybrids are designed to address multiple AD pathways, such as reducing Aβ aggregation, inhibiting tau hyperphosphorylation, enhancing antioxidant defenses, and modulating neurotransmitter systems. By targeting various mechanisms, they offer a more comprehensive treatment approach with potentially improved therapeutic outcomes(89, 90) as presented in Fig 4.“

Reviewer 1: 5. Novel Donepezil-based hybrids with a focus on N-benzylpiperidine derivatives for 242 targeting AD (2014-2024)

-          Lines no 244-250 Multi-target-directed ligands (MTDLs)….are the repetition of already listed definitions and as such is unnecessary. Suggestion is to start this part with the story about donepezil analogues…without repetiting alreday explained definitions

Response and Revisions: We removed alreday explained definitions of Multi-target-directed ligands (MTDLs)

We added the text: “Heterocyclic compounds are a key in MTDL development, requiring precise chemical synthesis to ensure efficacy and safety. The goal is to better address the complex nature of neurodegenerative diseases, potentially offering advantages over single-target therapies. Here we examine advancements in donepezil-based hybrids, focusing on the design, synthesis, and evaluation of N-benzylpiperidine derivatives for Alzheimer’s disease treatment over the past decade. The piperidine moiety can interact with various biological targets, making it a valuable scaffold in medicinal chemistry. N-Benzyl substitution increases lipophilicity, enhancing membrane penetration and bioavailability. The nitrogen atom in the piperidine ring adds electron density, influencing binding affinity to neuropharmacological receptors and enzymes. Additionally, the structure allows conformational flexibility, enabling optimal interactions with target proteins. This versatility aids in designing new derivatives with improved efficacy and selectivity while reducing side effects. The nitrogen atom can also participate in hydrogen bonding, further enhancing binding affinity to specific targets. This review highlights also the potential of various natural compounds as multi-targeted therapies for Alzheimer's disease (AD), addressing multiple disease pathways.”

Reviewer 1: -          Table 2. Novel donepezil-based hybrids is unreadable in this form, and repetitve. I suggest to author to put only the structures of the compounds and their targetets in the Table, because the explanation of every compound  is already in the text of the section.

Response and Revisions: Thanks for the suggestion. We have simplified the table by removing the explanations that are in the text.

Reviewer 1: -          Line no 307 is repetition…

Response and Revisions: We removed the sentence: Aβ1−42 and acetylcholinesterase (AChE) are critical therapeutic targets for Alzheimer's disease (AD).

Reviewer 1: -         The structure of compounds  3b, d5 and d10 are missing from the Table 2

Response and Revisions: We removed the compounds 3b, d5 and d10 from the table

Reviewer 1: 6. Natural compounds as multitarget drugs for Alzheimer's disease treatment

- once again, there is only one reference at the begining of this section

 Response and Revisions: We have added references

Reviewer 1: Conclusion

Emphasize the contributions your literature review makes to the field. Discuss how your review  sheds new light on existing knowledge.

Response and Revisions: The conclusion has been revised and highlighted in red within the text for clarity.

 Reviewer 1: Reference

There is several references that are listed more than once.

Response and Revisions: The references have been revised.

Reviewer 2 Report

Comments and Suggestions for Authors

The authors are kindly requested to address the following major critical issues:

Title: It is too generic and not attractive. There  several reviews having the same title! Since the review is mainly focused on natural compounds as multitarget drugs for Alzheimer's disease treatment, the Title should be revised accordingly.

Important and recent contributions exploiting natural compounds surprisingly have been omitted:

Few examples:

i) juglone hybrids - see: doi: 10.1021/jm5010804

ii) licochalcone A - see: doi:10.3390/ijms241814177

iii) alkaloids hybrids - see: doi:10.3390/ijms24054399

iv) Natural products - see doi.10.3389/fnins.2022.884345

v) AChE-BChE- BACE1 multi targets should be better addressed. See: 10.1016/j.ejmech.2023.115253 and doi: 10.3390/ph16121657

Minor Issues:

The authors are kindly request to validate the spelling of the compounds.

There are several typos throughout the entire manuscript.

One Example:

Figure 2

"Ravastigmine" should read "Rivastigmine"

"Mamantine" should read "Memantine".

References

Some of the references have been reported in duplicate and even triplicates!

See references: 48, 50, and 55

Author Response

Response to Reviewer 2 Comments

Thank you for taking the time to review this manuscript. We have closely followed the reviewers' instructions and are pleased to submit the revised version. Below are our detailed responses, and the corresponding corrections are highlighted in track changes in the resubmitted file.

Comments and Suggestions for Authors

Reviewer 2: The authors are kindly requested to address the following major critical issues:

Title: It is too generic and not attractive. There  several reviews having the same title! Since the review is mainly focused on natural compounds as multitarget drugs for Alzheimer's disease treatment, the Title should be revised accordingly.

Response and Revisions: Thanks for the suggestion. We have changed the title of the manuscript as follows:

 "New Insights into the Development of Donepezil-Based Hybrid and Natural Molecules as Multi-Target Drug Agents for Alzheimer's Disease Treatment"

Reviewer 2: Important and recent contributions exploiting natural compounds surprisingly have been omitted:

Few examples:

  1. i) juglone hybrids - see: doi: 10.1021/jm5010804
  2. ii) licochalcone A - see: doi:10.3390/ijms241814177

iii) alkaloids hybrids - see: doi:10.3390/ijms24054399

  1. iv) Natural products - see doi.10.3389/fnins.2022.884345
  2. v) AChE-BChE- BACE1 multi targets should be better addressed. See: 10.1016/j.ejmech.2023.115253 and doi: 10.3390/ph16121657

Response and Revisions: Thanks for the suggestion. This manuscript focuses on key natural products primarily involved in clinical trials. To maintain clarity and conciseness, we have added only selected suggestions related to specific structures, marked in red color.

Reviewer 2: Minor Issues:

The authors are kindly request to validate the spelling of the compounds.

There are several typos throughout the entire manuscript.

One Example: Figure 2

"Ravastigmine" should read "Rivastigmine"

"Mamantine" should read "Memantine".

Response and Revisions: Thanks for the note. Errors have been corrected in the text and table in Fig 2.

Reviewer 2: References

Some of the references have been reported in duplicate and even triplicates!

See references: 48, 50, and 55

Response and Revisions: We apologize for the oversight, but we inadvertently did not submit the version with the finalized formatting of the references. In the current version, references have been added both in the introduction, from paragraph 40 to 65, and duplicate citations of authors have been avoided.

Round 2

Reviewer 1 Report

Comments and Suggestions for Authors

The manuscript has been improved by the authors and can be recommended for publication.

Author Response

Reviewer 1 comments: The manuscript has been improved by the authors and can be recommended for publication.

Response: Thank you very much for taking the time to review this manuscript.

Reviewer 2 Report

Comments and Suggestions for Authors

Overall, the authors have in part addressed the issues raised by the Reviewer.

The focus of the manuscript has been improved. 

Minor Issue:  Please validate the correct spelling throughout the manuscript.

See, as example:

3. Aduvants of multy-target drugs in Alzheimer's

should read

3. Adjuvants of multi-target drugs in Alzheimer's

Author Response

Reviewer 2 comments: Overall, the authors have in part addressed the issues raised by the Reviewer.

The focus of the manuscript has been improved.

Minor Issue:  Please validate the correct spelling throughout the manuscript.

Response: The spelling was validated throughout the MS

See, as example:

  1. Aduvants of multy-target drugs in Alzheimer's

should read

  1. Adjuvants of multi-target drugs in Alzheimer's

Response: It was done!